# Online Fast Adaptation and Knowledge Accumulation (OSAKA): a New Approach to Continual Learning

**Massimo Caccia**[123]    **Pau Rodríguez**[2]    **Oleksiy Ostapenko**[13]    **Fabrice Normandin**[13]
**Min Lin**[13]    **Lucas Caccia**[145]    **Issam Laradji**[2]    **Irina Rish**[137]
**Alexandre Lacoste**[2]    **David Vazquez**[2]    **Laurent Charlin**[167]

[1]Mila - Quebec AI Institute, [2]ElementAI, [3]Université de Montréal, [4]Facebook AI Research
[5]McGill University, [6]HEC Montréal, [7]Canada CIFAR AI Chair

## Abstract

Continual learning agents experience a stream of (related) tasks. The main challenge is that the agent must not forget previous tasks and also adapt to novel tasks in the stream. We are interested in the intersection of two recent continual-learning scenarios. In *meta-continual learning*, the model is pre-trained using meta-learning to minimize catastrophic forgetting of previous tasks. In *continual-meta learning*, the aim is to train agents for *faster remembering* of previous tasks through adaptation. In their original formulations, both methods have limitations. We stand on their shoulders to propose a more general scenario, OSAKA, where an agent must quickly solve new (out-of-distribution) tasks, while also requiring fast remembering. We show that current continual learning, meta-learning, meta-continual learning, and continual-meta learning techniques fail in this new scenario. We propose *Continual-MAML*, an online extension of the popular MAML algorithm as a strong baseline for this scenario. We show in an empirical study that *Continual-MAML* is better suited to the new scenario than the aforementioned methodologies including standard continual learning and meta-learning approaches.

## 1 Introduction

A common assumption in supervised machine learning is that the data is independently and identically distributed (i.i.d.). This assumption is violated in many practical applications handling non-stationary data distributions, including robotics, autonomous driving, conversational agents, and other real-time applications. Over the last few years, several methodologies study learning from non-i.i.d. data. We focus on *continual learning* (CL), where the goal is to learn incrementally from a non-stationary data sequence involving different datasets or *tasks*, while not forgetting previously acquired knowledge, a problem known as *catastrophic forgetting* [47].

We draw inspiration from autonomous systems deployed in environments that might differ from the ones they were (pre-)trained on. For instance, a robot pre-trained in a factory and deployed in homes where it will need to adapt to new domains and even solve new tasks. Or a virtual assistant can be pre-trained on historical data and then adapt to its user's needs and preferences once deployed. Further motivating applications exist in time-series forecasting including market prediction, game playing, autonomous customer service, recommendation systems, and autonomous driving. These systems must adapt online to maximize their cumulative rewards [30, 31]. As a step in that direction, we propose a task-incremental scenario (OSAKA) where previous tasks reoccur and new tasks appear.

---

corresponding author: massimo.p.caccia@gmail.com

We measure the cumulative accuracy of models instead of the (more common) final accuracy to evaluate how quickly models and algorithms adapt to new tasks and remember previous ones.

**Background**    *Task-incremental classification* is a common supervised CL scenario where classification datasets are presented to an online learner sequentially, one task at a time. For each task $T_t$ at iteration $t$, the data are sampled i.i.d. from their corresponding distribution $P_t(\mathbf{x}, \mathbf{y})$. In the task-incremental scenario models are evaluated by their average final performance across all tasks—after being trained on all tasks sequentially. Several families of recent CL approaches use this setting, including regularization methods [35], data replay [71], and dynamic architectures [64] (see Lange et al. [38] and Parisi et al. [54] for comprehensive overviews).

More recent approaches propose relaxing some constraints associated with task-incremental CL by combining CL and meta-learning. *Continual-meta learning* (CML) focuses on *fast remembering* or how quickly the model recovers its original performance on past tasks [24]. *Meta-continual learning* (MCL) uses meta-learning to learn not to forget [28]. In this paper, we further extend the task-incremental setting and show empirical benefits compared to CML and MCL (see Section 6).

**OSAKA**    We propose a more flexible and general scenario inspired by a pre-trained agent that must keep on learning new tasks after deployment. In this scenario, we are interested in the cumulative performance of the agent throughout its lifetime [30, 31]. (Standard CL reports the final performance of the agent on all tasks at the end of its *life*.) To succeed in this scenario, agents need the ability to learn new tasks as well as quickly remember old ones.

We name our CL setting *Online faSt Adaptation and Knowledge Accumulation* (OSAKA). The main characteristics of OSAKA are that at deployment or CL time: 1) task shifts are sampled stochastically, 2) the task boundaries are unknown (*task-agnostic* setting), 3) the target distribution is context-dependent, 4) multiple levels of non-stationarity are used, and 5) tasks can be revisited. Furthermore, our evaluation of CL performance is different from the one commonly used in CL. We report the cumulative or online average performance instead of the final performance on all seen tasks.

Existing CL methods are not well-suited to OSAKA. Methods such as EWC [35], progressive networks [64] or MCL [28] require task boundaries. In contrast, task-agnostic methods (e.g. [1, 83, 24]) optimize for the final performance of the model and so resort to mechanisms that attempt to eliminate catastrophic forgetting. The extra computations resulting from the mechanisms hinder online performance and unnecessarily increase the computational footprint of the algorithms.

To address the challenges of OSAKA, we propose *Continual-MAML*, a baseline inspired by the meta-learning approach of MAML [15]. Continual-MAML is pre-trained via meta-learning. When deployed, Continual-MAML adapts the learned parameter initialization to solve new tasks. When a change in the distribution is detected, new knowledge is added into the learned initialization. As a result, Continual-MAML is more efficient and robust to distribution changes since it does not require computationally expensive optimizers like BGD [83] or replay methods used in prior work [10, 68].

Using our OSAKA scenario, we compare the performance of Continual-MAML to recent and popular approaches from continual learning, meta-learning, and continual-meta learning. Across several datasets, we observe that Continual-MAML is better suited to OSAKA than prior methods from the aforementioned fields and thus provides an initial strong baseline.

To summarize, our contributions include: (1) OSAKA, a new CL setting which is more flexible and general than previous ones. Related, we also propose a unifying scenario for discussing meta- and continual learning scenarios (Table 4); (2) the Continual-MAML algorithm, a new baseline that addresses the challenges of the OSAKA setting; (3) extensive empirical evaluation of our proposed method; and (4) a codebase for researchers to test their methods in the OSAKA scenario.[1]

## 2   A unifying framework

We introduce the concepts and accompanying notation that we will use to describe OSAKA in Section 3. These concepts provide a unifying framework—highlighted in Table 4—for expressing several important paradigms such as continual learning, meta-learning, and variants. A motivation for this framework (and so this section in our paper) is to clarify some confusion that arose from the

| | Data Distribution | Model for Fast Weights | Slow Weights Updates | Evaluation |
|---|---|---|---|---|
| Supervised Learning | $S, Q \sim C$ | $f_\theta = \mathcal{A}(S)$ | — | $\mathcal{L}(f_\theta, Q)$ |
| Meta-learning | $\{C_i\}_{i=1}^M \sim \mathcal{W}^M$ $S_i, Q_i \sim C_i$ | $f_{\theta_i} = \mathcal{A}_\phi(S_i)$ | $\nabla_\phi \mathcal{L}(f_{\theta_i}, Q_i)$ $\forall i < N$ | $\sum_{i=N}^M \mathcal{L}(\mathcal{A}_\phi(S_i), Q_i)$ |
| Continual Learning | $S_{1:T}, Q_{1:T} \sim C_{1:T}$ | $f_\theta = \mathrm{CL}(S_{1:T})$ | — | $\sum_t \mathcal{L}(f_\theta, Q_t)$ |
| Meta-Continual Learning | $\{C_{i,1:T}\}_{i=1}^M \sim \mathcal{W}^M$ $S_{i,1:T}, Q_{i,1:T} \sim C_{i,1:T}$ | $f_{\theta_i} = \mathrm{CL}_\phi(S_{i,1:T})$ | $\nabla_\phi \sum_t \mathcal{L}(f_{\theta_i}, Q_{i,t})$ $\forall i < N$ | $\sum_{i=N}^M \sum_t \mathcal{L}(\mathcal{A}_\phi(S_{i,1:T}), Q_{i,t})$ |
| Continual-meta learning | $S_{1:T}, Q_{1:T} \sim C_{1:T}$ | $f_{\theta_t} = \mathcal{A}_\phi(S_{t-1})$ | $\nabla_\phi \mathcal{L}(f_{\theta_t}, S_t)$ | $\sum_t \mathcal{L}(\mathcal{A}_\phi(S_t), Q_t)$ |
| OSAKA | $Q_{1:T} \sim C_{1:T}$ | $f_{\theta_t} = \mathcal{A}_\phi(Q_{t-1})$ | $\nabla_\phi \mathcal{L}(f_{\theta_t}, Q_t)$ | $\sum_t \mathcal{L}(f_{\theta_t}, Q_t)$ |

Table 1: **A unifying framework** for different machine learning settings. Data sampling, fast weights computation and slow weights updates as well as evaluation protocol are presented with meta-learning terminology, i.e., the support set $S$ and query set $Q$. For readability, we omit OSAKA pre-training.

recent interrelation of meta-learning and continual learning. Our main contribution, OSAKA, can be understood even if the reader chooses to skip this section.

We begin by assuming a hidden context variable $C$ that determines the data distribution, e.g., the user's mood in a recommender system or an opponent's strategy in game playing. In some fields, contexts are referred to as *tasks*. In the rest of the paper, we will use both terms interchangeably. We use $\mathcal{W}$ to denote a finite set of all possible contexts. Given $C$, data can be sampled i.i.d. from $p(X|C)$. Different learning paradigms can be described by specializing the distribution $P(C)$. For example, in the classical setting data are sampled i.i.d. from $p(X|C)P(C)$ where $C$ could represent the set of classes to be discriminated.

We use terminology from meta-learning and introduce a *support set* $S$ and a *query set* $Q$ to denote the meta-training and meta-test sets [78], respectively. These sets are usually composed of $n$ i.i.d. samples $X_i = (\boldsymbol{x}_i, \boldsymbol{y}_i)$, generated conditionally from the context $C_i$. In some paradigms, including supervised learning, the target distribution is fixed, i.e. $p(\boldsymbol{y}|\boldsymbol{x}) = p(\boldsymbol{y}|\boldsymbol{x}, C)$. We refer to the setting where the equality does not hold as having *context-dependent targets*. We define a learning algorithm $\mathcal{A}$ as a functional taking $S$ as input and returning a predictor $f_\theta$, with $\theta$ parameters describing the behavior of the predictor, i.e. $f_\theta = \mathcal{A}(S)$. We also define a loss function $\mathcal{L}(f_\theta, Q)$ to evaluate the predictor $f_\theta$ on the query set $Q$.

In **meta-learning**, $C$ represents a task descriptor or task label, and both meta-training and meta-testing sets are sampled i.i.d. from $p(X|C)$. E.g., in $N$-shot classification, the task descriptor specifies the $N$ classes which have to be discriminated. Targets are context-dependent in this learning paradigm. Here, we focus only on the meta-learning methods that rely on episodic training.

A meta-learning algorithm $\mathcal{A}_\phi$ adapts its behavior by learning the parameters $\phi$. It samples $M$ i.i.d. pairs of $S$ and $Q$ from a distribution over contexts $\mathcal{W}^M$: $\{C_i\}_{i=1}^M \sim \mathcal{W}^M$ and $(S_i, Q_i) \sim X_i \mid C_i$. Assuming that the learning process is differentiable, the parameters $\phi$ are learned using the gradient from the query set, $\nabla_\phi \mathcal{L}(\mathcal{A}_\phi(S_i), Q_i)$. Concretely, $\phi$ is first learned on the sets $(S_i, Q_i)$, where $i < N < M$ and the final evaluation of the algorithm is $\sum_{i=N}^M \mathcal{L}(\mathcal{A}_\phi(S_i), Q_i)$.

In task-incremental **continual learning**, the data distribution is non-stationary, and various CL scenarios arise from specific assumptions about this non-stationarity. Here we assume that data non-stationarity is caused by a hidden process $\{C_t\}_{t=1}^T$, where $C_t$ is the context at time $t$. $C$ in continual learning can be the task label, e.g., in Permuted MNIST, disjoint MNIST/CIFAR10 [35]. It could also be the class label in the class-incremental setting [59]. Both frameworks have a fixed target distribution. $\{C_t\}_{t=1}^T$ is usually assumed to be an ordered list of the tasks/classes.

Continual learning algorithms work with a sequence of support sets, $S_{1:T}$, and a sequence of query sets, $Q_{1:T}$, obtained from a sequence of contexts, $C_{1:T}$. A continual learning algorithm CL transforms $S_{1:T}$ into a predictor $f_\theta$, i.e. $f_\theta = \mathrm{CL}(S_{1:T})$. The main difference with a conventional algorithm $\mathcal{A}$ is that the support set is observed sequentially and cannot be fully stored in memory. The evaluation is then performed independently on each $Q_t$ (obtained in the same context as $S_t$): $\sum_{t=1}^T \mathcal{L}(f_\theta, Q_t)$. In App. A we explain how the recent meta-continual learning and continual-meta learning settings fit into the unifying framework.

# 3 Online FaSt Adaptation and Knowledge Accumulation (OSAKA)

We propose OSAKA, a new continual-learning scenario that lifts some of constraints of current task-incremental approaches [35, 28, 2]. OSAKA is aligned with the use case of deploying a pre-trained agent in the real world, where it is crucial for the agent to adapt quickly to new situations and even to learn new concepts when needed. In particular, OSAKA proposes a scenario for evaluating such continually-learning agents.

To materialize such an evaluation OSAKA combines different ideas: 1) agents start in a pre-training stage before continual-learning starts; 2) it provides a mechanism for proposing both old and new tasks to agents where the task boundaries remain unobserved to them; 3) it evaluates agents using their cumulative performance (e.g. accuracy) to measure their capacity to adapt to new tasks. This evaluation implicitly allows agents to forget which may enable faster and more efficient adaptation. For instance, partially forgetting an infrequent task allows the agent to re-allocate modeling capacity to tasks that are encountered more frequently.

We now describe OSAKA using the procedural view of Alg. 1. OSAKA proposes a two-stage approach where an agent $\theta_0$ starts in a *pre-training phase* (Alg. 1, L4–L8) and then moves to a *deployment phase* (Alg. 1, L10–L16) also known as continual-learning time.

---

**Algorithm 1: OSAKA**

1 **Require:** $P(C_{\text{pre}})$, $P(C_{\text{cl}})$: distributions of contexts
2 **Require:** $\alpha$: non-stationarity level
3 **Initialize:** $\theta_0$: Model
4 **while** *pre-training*
5     Sample a context $C \sim P(C_{\text{pre}})$
6     Sample data from context $\boldsymbol{x}, \boldsymbol{y} \sim p(\boldsymbol{x}, \boldsymbol{y}|C)$
7     Update $\theta_0$ with $\boldsymbol{x}, \boldsymbol{y}$
8 **end**
9 **while** *continually learning*
10     Sample current context $C_t \sim P(C_{\text{cl}}|C_{t-1}; \alpha)$
11     Sample data from context $\boldsymbol{x}_t, \boldsymbol{y}_t \sim p(\boldsymbol{x}, \boldsymbol{y}|C_t)$
12     Incur loss $\mathcal{L}\big(\theta_{t-1}(\boldsymbol{x}_t), \boldsymbol{y}_t\big)$
13     Update $\theta_t$ with $\boldsymbol{x}_t, \boldsymbol{y}_t$ at discretion
14     $t \leftarrow t + 1$
15 **end**

**Algorithm 2: Continual-MAML at CL time**

1 **Require:** $\eta, \gamma, \lambda$: learning rate, hyperparameters
15 **while** *continually learning*
16     $C_t \sim P(C_{\text{cl}}|C_{t-1})$
17     $\boldsymbol{x}_t, \boldsymbol{y}_t \sim P(\boldsymbol{x}, \boldsymbol{y}|C_t)$
18     $\mathcal{L}\big(f_{\theta_{t-1}}(\boldsymbol{x}_t), \boldsymbol{y}_t\big)$
19     $\tilde{\theta}_t \leftarrow \phi - \phi_\eta \nabla_\phi \mathcal{L}\big(f_\phi(\boldsymbol{x}_t), \boldsymbol{y}_t\big)$
20     **if** $\mathcal{L}\big(f_{\theta_{t-1}}(\boldsymbol{x}_t), \boldsymbol{y}_t\big) - \mathcal{L}\big(f_{\tilde{\theta}_t}(\boldsymbol{x}_t), \boldsymbol{y}_t\big) < \gamma$
21         $\theta_t \leftarrow \theta_{t-1} - \phi_\eta \nabla_\theta \mathcal{L}\big(f_{\theta_{t-1}}(\boldsymbol{x}_t), \boldsymbol{y}_t\big)$
22     **else**
23         $\eta_t \leftarrow \eta g_\lambda \big(\mathcal{L}\big(f_{\theta_{t-2}}(\boldsymbol{x}_{t-1}), \boldsymbol{y}_{t-1}\big)\big)$
24         $\phi \leftarrow \phi - \eta_t \nabla_\phi \mathcal{L}\big(f_{\theta_{t-2}}(\boldsymbol{x}_{t-1}), \boldsymbol{y}_{t-1}\big)$
25         $\theta_t \leftarrow \phi - \phi_\eta \nabla_\phi \mathcal{L}\big(f_\phi(\boldsymbol{x}_t), \boldsymbol{y}_t\big)$
26     $t \leftarrow t + 1$
27 **end**

---

**Pre-training (Alg. 1 L4–L8).** In many current settings [35, 24], the agent begins learning from randomly-initialized parameters. However, in many scenarios, it is unrealistic to deploy an agent without any world knowledge [42, 44], in part, since real-life non-i.i.d. training is difficult to learn. Further, in many domains, ample pre-training data can be leveraged.

**Continual-learning time (Alg. 1 L9–L15)** After pre-training, a stream of continual learning tasks evaluate the model. Each iteration $t$ in the stream relies on a context $C_t$ which determines the current task $(\mathbf{x}_t, \mathbf{y}_t)$. The contexts follow a Markov process $\{C_t\}_{t=1}^T$ with transition probabilities $P(C_t|C_{t-1}; \alpha)$ (Alg. 1, L10).

The context is at the heart of OSAKA and its process controls the level of stationarity of the continual-learning stage and it enables both revisiting tasks and out-of-distribution ones as well as context-dependent targets. We discuss these features below.

**Controllable non-stationarity.** OSAKA provides control, through a hyperparameter, over the level of non-stationarity of the Markov chain. A stream is $\alpha$-locally-stationary when $P(C_t = c|C_{t-1} = c) = \alpha$. Namely, the data distribution is stationary within a local-time window, i.e., over a certain amount of timesteps. Control over $\alpha$ enables exploring environments with different levels of non-stationarity to test algorithmic robustness.

Similar to the few-shot learning literature [78, 58, 52, 62], the transitions of the context variables in OSAKA are not structured, i.e. the context transition matrix that encodes the probability of transitioning from context $i$ to context $j$ has $\alpha$ on the diagonal and $(1 - \alpha)/(|C| - 1)$ everywhere else. For that reason, modelling the evolution of the context variables is not essential. Further, in OSAKA the environment provides enough feedback to the agents for re-adaptation via the targets

$\boldsymbol{y}_t$ (Alg. 1, L13). We leave the design of a continual learning experimental setup and associated modeling with a structured context variable for future work.

**Task revisiting.** Standard CL methods incrementally learn strictly new tasks. However, many CL applications require revisiting previous tasks. Through the process $\{C_t\}_{t=1}^T$ OSAKA proposes task revisiting, analogous to *recurrent concept drift* [17] in online learning. By revisiting previous data distributions, methods will enjoy the same form of implicit replay that agents and systems naturally benefit from in real-world scenarios. The domain of each context $C_t$ contains all tasks and so the process allows to switch back and forth from old tasks to OoD tasks.

**Out-of-distribution (OoD) tasks.** Current settings that permit pre-training then continually learn tasks sampled from the same data distribution [28, 6]. In contrast, in OSAKA the model has to learn online tasks sampled from new distributions not encountered at pre-training (see Section 6.1 for details). This setting is more realistic since an agent will encounter unexpected situations in real life requiring the algorithm to update its representations.

**Context-dependent targets.** In standard CL, $p_t(\boldsymbol{x})$ shifts over time, but the target distribution $p(\boldsymbol{y}|\boldsymbol{x})$ is fixed. However, drift in the target distribution is common in multiple applications and is studied extensively in online learning as *real concept drift* [17]. Extending [24], OSAKA allows for context-dependent targets (Alg. 1, L11) making it more flexible and more aligned with our use-cases (see Sec. 1). In OSAKA the target distribution is $p(\boldsymbol{y}|\boldsymbol{x}, C_t)$.

The context variable in OSAKA is generic but it is motivated by real-world domains. For example, the context variable could be the strategy of an opponent in a game [69, 49, 77, 51], regimes in time-series forecasting [56, 19, 9], the mood of a user when navigating a content platform in recommender systems [25, 72] or any unobserved variable in RL, e.g., in partially observable Markov decision processes (POMDPs) [32] or in hidden-mode Markov decision processes [12]. In all these examples, like in OSAKA, the targets change over time based on a context.

**Task agnostic.** In OSAKA the agent does not observe the task boundaries or context shifts, and it must infer the current task or context $C_t$. This is called task-agnostic (or task-free) CL [3, 4, 83, 24, 42] and is motivated by real-world scenarios where signals explicitly indicating a shift may not exist.

**Online Evaluation (Alg. 1 L12).** Current settings reward methods that retain their performance on all previously seen tasks. This is an unrealistic constraint, particularly under limited computational resources [30, 31]. Instead of measuring the final performance of the model, OSAKA measures the online cumulative performance which better suits non-stationary environments. Models are evaluated in an online fashion using the sum of the losses across all timesteps $\sum_{t=1}^T \mathcal{L}(f_{\theta_t}, Q_t)$ where $\mathcal{L}$ can be any loss (Alg. 1, L12). This is as opposed to reporting only the final accuracy—for example $\sum_{t=1}^T \mathcal{L}(f_{\theta_T}, Q_t)$ [35, 59, 10, 11, 28]. Similar to the final accuracy, the online cumulative accuracy measures both plasticity and stability. Specifically, plasticity is evaluated when the algorithm encounters OoD tasks requiring additional learning. Models with higher stability can recover past performance faster and thus enjoy higher online cumulative performance. The cumulative accuracy is also similar to evaluating the (undiscounted) sum of rewards in reinforcement learning or the regret [7] in online learning albeit without the need to compute the performance of an optimal model.

We instantiate OSAKA for image classification tasks (see Sec. 6), similarly to the majority of CL benchmarks. Some motivations for our proposed experimental setting are drawn, however, from a reinforcement learning (RL) scenario. In fact, we could adapt OSAKA to RL. We could replace the image classification tasks by tasks from multi-task RL benchmarks (e.g., such as different robotic tasks [82, 27]). We could also use a standard RL benchmark, e.g, from a model-based control environment [75], and create different contexts by changing some of the environment variables, e.g. the gravity. Once tasks or contexts are defined, we could group them in such a way that the CL-time tasks are OoD with respect to pre-training tasks. Increasingly more complex tasks could also be introduced at CL time to mimic a curriculum-learning scenario. Finally, to control for different levels of non-stationarity, we could adjust the time allocation or the number of episodes in each context/task.

## 4 Continual-MAML

We propose Continual-MAML (see Fig. 1), a CL baseline based on MAML [15] that can cope with the challenges of OSAKA. Continual-MAML (see Alg. 2 or its complete version Alg. 3 in App. B) consists of two stages: pre-training and continual learning.

The pre-training phase consists of MAML. That is, meta-learning model parameters such that a small number of gradient steps on a small new task will produce good generalization performance on that task (Alg. 3, L6–13). Specifically, the model adapts its initial weights $\phi$ to multiple tasks in the inner loop, obtaining $\theta$. Then it updates the initialization $\phi$ in the outer loop. Note that the inner loop learning rate is meta-learned ($\phi_\eta$ in Alg. 3, L10).

At CL time (Alg. 2), the inner loop optimization adapts the model to the current task. Specifically, the model uses current data $X_t, Y_t$ to obtain fast weights $\theta_t$ (Alg. 2, L21). Assuming that the data is locally stationary, it makes a prediction on the following data $X_{t+1}$ and incurs a loss (Alg. 2, L18). In the case of a sudden distribution shift, the model will fail at its first prediction because its fast weights $\theta_t$ are not suited for the new task yet, but it will have recovered by the next. The recovery is achieved by learning new fast weights $\theta_{t+1}$ once the algorithm gets feedback on its prediction (Alg. 2, L25). Note that for some real-life applications, this feedback could be delayed [33]. Finally, to accumulate new knowledge, we further update the meta parameters $\phi$ on the incoming data as well (Alg. 2, L24).

We also propose two features to improve Continual-MAML's performance. First, the algorithm must update its knowledge only when it is solving an OoD task. Accordingly, we introduce a hyperparameter $\lambda$ that controls the behavior of the algorithm between never training on the incoming data at CL time to always training (MAML and C-MAML in Section 6). Specifically, when $\mathcal{L}(f_{\theta_{t-1}}(X_t), Y_t) > \lambda$, new knowledge is incorporated through outer loop optimization of the learned initialization. This mechanism is exemplified in Figure 1. To obtain a smoother interpolation between behaviors, we opted for a soft relaxation of the mechanism (Alg. 2, L23) where $g_\lambda : \mathbb{R} \to (0, 1)$. We call this first feature *update modulation* (UM).

Second, to further leverage the local stationarity of OSAKA, we introduced a mechanism that keeps fine-tuning the fast weights $\theta$ (Alg. 2, L21) until a context shift or task boundary is detected. The simple yet effective context shift detection mechanism works by monitoring the difference in loss with respect to the previous task and is controlled by a hyperparameter $\gamma$ (Alg. 2, L20). We call this second feature *prolonged adaptation phase* (PAP). In practice, we use a buffer to accumulate data whilst no task boundary is detected such that we can update the slow weights $\phi$ with more examples once it's detected (see Alg. 3 in App. B). One can think of the update after the task boundary detection as a knowledge consolidation phase.

An ablation of both mechanisms and an hyperparameter sensitivity analysis are provided in Section 6.3 and Appendix F.2, respectively.

As a result, different from previous CL literature, the proposed algorithm benefits from fast adaptation, dynamic representations, task boundary detection, and computational efficiency, as we describe next.

**Fast Adaption**   During pre-training, Continual-MAML learns a weight initialization that adapts fast to new tasks. This is different from CL methods that focus on incorporating as much knowledge as possible into one representation that has to maximize performance in a multi-task regime.

**Dynamic representations**   In OSAKA, significant distribution shifts occur periodically. As shown in Section 6, models that require a fixed representation would fail to adapt. Instead, Continual-MAML, equipped with UM, detects OoD data and then learns new knowledge using outer-loop optimization.

**Computational efficiency**   As described by Farquhar and Gal [14], CL agents should operate under restricted computational resources since remembering becomes trivial in the infinite-resource setting.

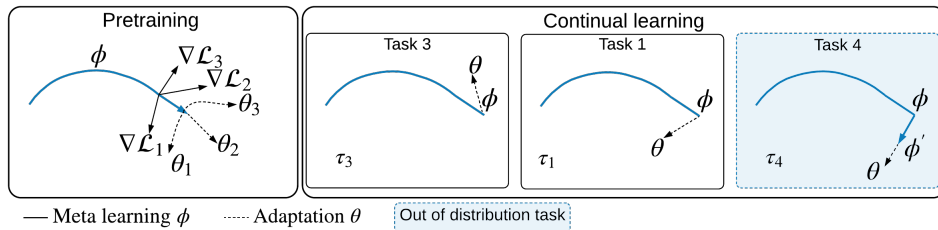

Figure 1: **Continual-MAML** first pre-trains with MAML, obtaining $\phi$. At continual-learning time, the model adapts $\phi$ to new distributions. The algorithm retrains its slow weights $\phi$ when it detects an OoD task to add new knowledge to the model. (Figure is adapted from Figure 1 in Finn et al. [15].)

Continual-MAML satisfies this desideratum by allowing the agent to forget (to some extent) and re-allocate parametric capacity to new tasks. Likewise, no computationally expensive mechanisms, such as replay [11], or BGD [83, 24], are used to alleviate catastrophic forgetting in our method.

**Task boundary detection**  Continual-MAML detects context shifts which not only help to condition its predictive function on more datapoints (PAP), it also avoids mixing gradient information from two different distributions.

# 5   Related Work

Continual Learning (CL) [47, 74] has evolved towards increasingly challenging and more realistic evaluation protocols. The first evaluation frameworks [20, 35] were made more general in [83, 3] via the removal of the known task boundaries assumption. Later, [24] proposed to move the focus towards *faster remembering*, or continual-meta learning (CML), which measures how quickly models recover performance rather than measuring the models' performance without any adaptation. OSAKA builds upon this framework to get closer to real-life applications of CL, as explained in Section 3.

Harrison et al. [23] propose a new CML framework and accompanying model (MOCA). OSAKA shares commonalities with this framework, but they are fundamentally different: it does not (1) allow context-dependent targets, (2) expose the algorithms to OoD tasks at CL time, (3) allow new unknown labels, nor (4) propose an update CL evaluation protocol. Further details are in Appendix C.1.

# 6   Experiments

We study the performance of different baselines in the proposed OSAKA setup. We first introduce the datasets, methods, and baselines, and then report and discuss experimental results and observations.

## 6.1   Experimental setup

For all datasets we study two different levels of non-stationarity at CL time, $\alpha$ values of 0.98 and 0.90. Unless otherwise stated the continual-learning episodes have a length of 10,000 timesteps; the probability to visit the pre-training distribution and to visit one of the OoD ones is 0.5 and 0.25, respectively; we report the performance averaged over 20 runs per model and their standard deviation. Statistical significance is assessed using a 95% confidence interval and highlighted in bold. Further experimental details are provided in Appendix E. We now introduce our three datasets. A few examples from each are shown in Appendix D.

**Omniglot / MNIST / FashionMNIST**  In this study, we pre-train models on the first 1,000 classes of Omniglot [37]. At CL time, the models are exposed to the full Onniglot dataset, and two out-of-distribution datasets: MNIST [39] and FashionMNIST [80]. Concerning the reported performance, MNIST is a simpler dataset than Omniglot, and FashionMNIST is the hardest. During CL time, the tasks switch with probability $1-\alpha$. For this study, we sample 10-way 1-shot classification tasks.

**Synbols**  In this study, models are pre-trained to classify characters from different alphabets on randomized backgrounds [36]. Tasks consist of 4 different symbols with 4 examples per symbol. During CL time, the model is exposed to a new alphabet. Further, the model will have to solve the OoD task of font classification, where the input distribution does not change, only its mapping to the output space. The font classification task consists of 4 different fonts with 4 symbols per font.

**Tiered-ImageNet**  Like Omniglot, Tiered-ImageNet [60] groups classes into super-categories corresponding to higher-level nodes in the ImageNet [13] hierarchy (we use 20/6/8 disjoint sets for training/validation/testing nodes). We use these higher-level splits to simulate a shift of distribution. We follow the original splits, where the test set contains data that is out of the training and validation distributions. Thus, we use their training set for pre-training, and introduce their validation and test sets at CL time. We refer to them as train, test and OoD in Table 3, respectively. Since only one of the two introduced sets is OoD, we increase its probability of being sampled to 0.5, in accordance to the previous benchmarks. This experiment uses 20,000 steps (twice as the others).

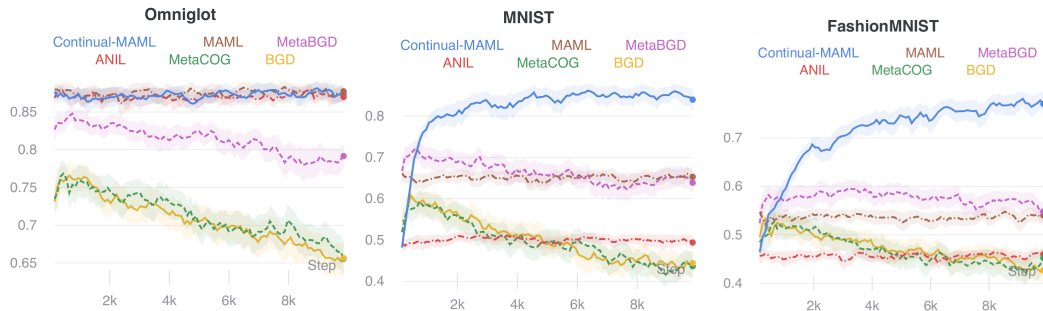

Figure 2: **Omniglot / MNIST / FashionMNIST experiment** in the $\alpha = 0.90$ regime. Methods are allowed pre-training on Omniglot before deployment on a stream of Omniglot, MNIST and FashionMNIST tasks. We report the online performance (not cumulative) at each time-step with averaged over 20 runs, as well as standard error. Online ADAM and Fine tuning lie below of the graph. Continual-MAML is the only method with enough plasticity to increase its performance on new tasks, i.e. from MNIST and FashionMNIST, whilst simultaneously being stable enough remember the pretraining tasks, i.e. from Omniglot.

## 6.2 Baselines

Appendix D compares the main features of the baselines we benchmark in the OSAKA setting. For meta-learning methods, ADAM [34] and SGD are used for the outer and inner updates, respectively.

**Online ADAM and Fine tuning.** We use ADAM without and with pre-training as a lower bounds.

**BGD [83].** Bayesian Gradient Descent (BGD) is a continual learning algorithm that models the distribution of the parameter vector $\phi$ with a factorized Gaussian. Similarly to [24] we apply BGD during the continual learning phase. More details about this baseline are provided in Appendix G.1.

**MAML [15].** MAML consists of a pre-training stage and a fine-tuning stage. During pre-training, the model learns a general representation that is common between the tasks. In the fine-tuning stage, the model fine-tunes its layers to adapt to a new task.

**ANIL [57].** ANIL differs from MAML only in the fine-tuning stage. Instead of adapting all the network layers, ANIL adapts only the network's head towards the new task. The goal of this baseline is to show the problem with static representations in the continual learning setup. Therefore, ANIL is representative of meta-continual learning.

**MetaBGD and MetaCOG [24].** MetaBGD performs CML using MAML and BGD to alleviate catastrophic forgetting. MetaCOG introduces a per-parameter mask learned in the inner loop.

## 6.3 Experimental results

For all benchmarks, we report results on two $\alpha$-locally-stationary environments. The first benchmark's results show online accuracy as function of timesteps in Figure 2 (full results are found in Appendix F.1). For Synbols and Tiered-Imagenet, the average accuracies over time are reported in Tables 2 and 3, respectively. For both regimes, the first column is the average performance over all predictions. The second, third and fourth columns show the performance on the three different settings. The prefix PRE. stands for pretraining. Algorithms perform better in the more locally-stationary regime ($\alpha = 0.98$) because they spend more time in each task before switching.

**Fast adaptation** We found fast adaptation (or meta-learning) to be the most critical feature for models to perform well in OSAKA, as highlighted by the performance gap between Online ADAM and Continual-MAML (up to +33% in Synbols $\alpha = 0.90$). This gain comes from two advantages: quickly changing weights after a task/context switch, having slow ($\phi$), and fast ($\theta$) weights, which alleviate catastrophic forgetting.

**Dynamic representations** Next, models need the ability to adapt the embedding space to correctly classify the OoD data. The Synbols font classification task highlights that learning a new mapping from the same inputs to a new output space is challenging when the embedded space is static.

| MODEL | α = 0.98 | | | | α = 0.90 | | | |
|---|---|---|---|---|---|---|---|---|
| | TOTAL | PREV. ALPH. | NEW ALPH. | FONT CLASS. | TOTAL | PREV. ALPH. | NEW ALPH. | FONT CLASS. |
| ONLINE ADAM | 59.6 ±1.5 | 63.7 ±2.3 | 59.5 ±3.7 | 50.7 ±2.9 | 27.5 ±0.8 | 28.3 ±1.1 | 26.3 ±0.9 | 26.9 ±0.7 |
| FINE TUNING | 64.0 ±2.0 | 69.6 ±2.1 | 63.0 ±3.6 | 52.9 ±2.8 | 26.6 ±1.8 | 27.0 ±2.4 | 26.2 ±1.5 | 26.1 ±1.2 |
| MAML [15] | 71.2 ±2.8 | 90.3 ±0.8 | 65.7 ±1.5 | 37.9 ±1.1 | 69.3 ±0.9 | 86.3 ±0.5 | 64.3 ±0.7 | 40.4 ±0.7 |
| ANIL [57] | 69.4 ±1.9 | 91.3 ±0.8 | 59.0 ±1.6 | 33.2 ±1.0 | 70.2 ±0.8 | **88.4** ±**0.4** | 68.7 ±0.6 | 35.1 ±0.5 |
| BGD [83] | 68.3 ±1.4 | 73.6 ±2.3 | 69.7 ±3.0 | 56.1 ±3.5 | 33.9 ±1.3 | 36.7 ±1.6 | 32.0 ±1.9 | 30.3 ±0.9 |
| METACOG [24] | 68.3 ±1.7 | 73.6 ±1.7 | 69.6 ±2.8 | 56.8 ±2.8 | 34.6 ±1.3 | 37.1 ±1.8 | 33.5 ±2.4 | 30.5 ±1.0 |
| METABGD [24] | 72.5 ±1.6 | 77.8 ±1.8 | 73.6 ±1.7 | 58.8 ±3.5 | 60.3 ±0.4 | 65.8 ±0.7 | 62.2 ±1.4 | 47.8 ±1.4 |
| C-MAML | 74.4 ±1.4 | 79.4 ±1.1 | 76.3 ±2.6 | 61.6 ±3.1 | 61.2 ±2.5 | 66.5 ±3.1 | 62.9 ±2.8 | 49.3 ±1.7 |
| C-MAML + PRE. | 78.4 ±1.0 | 86.6 ±1.0 | 78.2 ±1.4 | 60.9 ±2.6 | 73.3 ±1.2 | 82.0 ±1.1 | 75.0 ±1.6 | 53.8 ±1.5 |
| C-MAML + PRE. + UM | 74.8 ±4.0 | 81.6 ±6.2 | 75.5 ±4.5 | 59.5 ±3.2 | 72.8 ±0.9 | 81.4 ±1.2 | 74.4 ±1.3 | 54.4 ±1.6 |
| C-MAML + PRE. + UM+ PAP | **86.3** ±**0.8** | **93.4** ±**0.6** | **86.7** ±**1.8** | **72.0** ±**2.4** | **76.3** ±**0.8** | 84.9 ±0.7 | **76.4** ±**1.5** | **58.5** ±**1.4** |

Table 2: Online cumulative accuracy for the **Synbols experiments**. Methods are allowed character classification pre-training on an alphabet. Then, they are deployed on a stream of tasks sampled from the pre-training alphabet and a new alphabet, as well as a font classification tasks on the pre-training alphabet. Continual-MAML + pre. outperforms all others methods in total cumulative accuracy and the PAP further increases performance.

| MODEL | α = 0.98 | | | | α = 0.90 | | | |
|---|---|---|---|---|---|---|---|---|
| | TOTAL | TRAIN | TEST | OOD | TOTAL | TRAIN | TEST | OOD |
| ONLINE ADAM | 44.5 ±1.7 | 43.9 ±2.1 | 44.6 ±2.2 | 44.6 ±2.1 | 22.7 ±0.2 | 22.7 ±0.4 | 22.6 ±0.4 | 22.7 ±0.3 |
| FINE TUNING | 44.6 ±1.5 | 43.8 ±2.8 | 44.1 ±2.1 | 45.2 ±1.8 | 22.6 ±0.2 | 22.5 ±0.3 | 22.7 ±0.4 | 22.6 ±0.3 |
| MAML [15] | 59.3 ±1.2 | 61.4 ±1.9 | 61.0 ±1.8 | 57.3 ±1.0 | **60.4** ±**0.4** | **63.2** ±**0.7** | **62.6** ±**0.5** | 58.0 ±0.3 |
| ANIL [57] | 62.4 ±0.7 | 65.7 ±0.8 | 64.8 ±1.3 | 59.5 ±0.9 | 58.1 ±0.5 | 61.0 ±0.8 | 59.7 ±0.7 | 55.8 ±0.4 |
| BGD [83] | 54.8 ±0.8 | 53.8 ±1.0 | 54.6 ±1.9 | 55.3 ±1.0 | 27.7 ±0.7 | 27.4 ±0.7 | 27.7 ±0.8 | 27.8 ±0.8 |
| METACOG [24] | 55.2 ±0.7 | 54.1 ±1.1 | 55.8 ±1.6 | 55.4 ±1.0 | 24.5 ±0.2 | 23.9 ±0.4 | 24.0 ±0.3 | 25.1 ±0.3 |
| METABGD [24] | 55.9 ±0.6 | 55.7 ±0.9 | 54.1 ±1.4 | 56.8 ±0.9 | 46.8 ±0.8 | 45.8 ±1.1 | 46.8 ±1.0 | 47.3 ±0.9 |
| C-MAML | 61.4 ±0.5 | 59.5 ±1.4 | 61.2 ±1.3 | 62.4 ±0.9 | 53.7 ±0.3 | 52.0 ±0.6 | 53.0 ±0.6 | 54.9 ±0.5 |
| C-MAML + PRE. | 59.1 ±0.9 | 57.4 ±1.2 | 58.4 ±1.8 | 60.1 ±1.2 | 57.8 ±0.7 | 56.3 ±0.7 | 57.7 ±0.9 | 58.6 ±0.7 |
| C-MAML + PRE. + UM | 66.7 ±0.9 | 65.7 ±1.7 | 66.2 ±1.6 | 67.4 ±0.9 | 59.7 ±0.3 | 59.1 ±0.8 | 59.7 ±0.6 | **59.9** ±**0.4** |
| C-MAML + PRE. + UM + PAP | **69.1** ±**0.7** | **68.7** ±**0.9** | **69.3** ±**1.0** | **69.1** ±**1.2** | 53.4 ±6.4 | 53.5 ±6.1 | 53.7 ±6.2 | 53.2 ±6.6 |

Table 3: Online cumulative accuracy for the **Tiered Imagenet experiment** (see Sec. 6.1 for the experimental details). For this experiment, Continual-MAML outperforms others methods in the more non-stationary regime ($\alpha = 0.98$). However, in the less-nonstationary one, MAML achieves better results due to its higher stability. Additionally, the UM mechanism consistently improved Continual-MAML's performance.

Namely, the dynamic representations of Continual-MAML offer a 23.7% and a 28.4% improvement in $\alpha = 0.98$ compared to MAML and ANIL. This behavior is demonstrated in Figure 2 were these two baselines do not improve their performances over time, which is precisely the goal of CL. Thus, these results demonstrates the inapplicability of current MCL to real scenarios. Although MCL can continually learn new tasks without forgetting, its static embedded space will prevent it from learning tasks lying outside of the pre-training data distribution.

**Computational efficiency** Moreover, adding BGD to slow-down forgetting hinder the acquisition of new knowledge. Removing this feature, e.g. from MetaBGD to Continual-MAML, increases the performance in five out of six experiments and diminishes the computation cost by 80%.

**Update modulation** We now analyse, via ablations, the mechanisms we added to Continual-MAML for further improvements. Modulating the updates improved the performance in Omniglot and Tiered experiments but decreased it in Synbols' (C-MAML + PRE. vs. C-MAML + PRE. + UM, resulting in an average increase of 1.7%. In Appendix F.2, we show how this mechanism interpolates C-MAML + UM's behavior betweeen MAML and C-MAML.

**Prolonged adaptation phase** Finally, our PAP enabled by the task boundary detection mechanism helps achieve impressive gains in the locally more stationary regime (+11.5% and 2.4% in Synbols and Tiered-ImageNet, respectively). In the other regime ($\alpha = 0.90$), the shorter task sequences limits the room for improvements and the results are inconclusive. An hyperparameter sensitivity analysis on $\gamma$ (see App. F.2) in terms of precision and recall for boundary detection accuracy shows that difference in loss magnitudes (see Alg. 2 L20) is a good signal for detecting context shifts.

# 7 Conclusions

We propose OSAKA a new approach to continual learning that focuses on online adaptation, faster remembering and is aligned to real-life applications. This framework is task agnostic, allows context-conditioned targets and task revisiting. Furthermore, it allows pre-training, and introduces OoD tasks at continual-learning time. We show that the proposed setting is challenging for current methods that were not designed for OSAKA. We introduce Continual-MAML, an initial baseline that addresses the challenges of OSAKA and we empirically demonstrate its effectiveness.

## Broader Impact

Our work proposes a more-realistic (synthetic) continual-learning environment. This research could help accelerate the deployment of CL algorithms into applications such as autonomous driving, recommendation systems, information extraction, anomaly detection, and others. A domain often associated with continual learning is health care. In health care, patient data is (usually) very sensitive, CL algorithms can be the solution to accumulating knowledge from different hospitals: they can be trained continually across hospitals without the data ever leaving the premise.

**Possible negative impacts:** Our framework enables previous tasks to be forgotten at different rates. If data is patient-level data and different tasks relate to different subset of patients, then it means that the system's performance on past patients could vary. A diagnostic system, for example, could forget how to properly diagnose a patient from a previous population whilst learning about a new one. Further research, possibly at the intersection of continual learning and fairness, is needed before the safe deployment of these algorithms.

**Possible positive impacts:** The aforementioned negative impact may also be its greatest asset for having a positive impact. Returning to our example, practitioners could understand to what extent a diagnostic system forgets previous diagnostics. They could then use and develop OSAKA to calibrate their algorithms to match their desiderata (e.g., by choosing when the negative consequence of forgetting may outweight the benefits of additional training data).

## Acknowledgments and Disclosure of Funding

Laurent Charlin is supported through a CIFAR AI Chair and grants from NSERC, CIFAR, IVADO, Samsung, and Google. Massimo Caccia is also supported through a MITACS grant. We would like to thank Grace Abuhamad for an helpful discussion on broader impacts.

## Footnotes

[1]https://github.com/ElementAI/osaka

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
