[Supplementary Material]

# A   A unifying framework

| | Data Distribution | Model for Fast Weights | Slow Weights Updates | Evaluation |
|---|---|---|---|---|
| Supervised Learning | $S, Q \sim C$ | $f_\theta = \mathcal{A}(S)$ | — | $\mathcal{L}(f_\theta, Q)$ |
| Meta-learning | $\{C_i\}_{i=1}^M \sim \mathcal{W}^M$ <br> $S_i, Q_i \sim C_i$ | $f_{\theta_i} = \mathcal{A}_\phi(S_i)$ | $\nabla_\phi \mathcal{L}(f_{\theta_i}, Q_i)$ <br> $\forall i < N$ | $\sum_{i=N}^M \mathcal{L}(\mathcal{A}_\phi(S_i), Q_i)$ |
| Continual Learning | $S_{1:T}, Q_{1:T} \sim C_{1:T}$ | $f_\theta = \mathrm{CL}(S_{1:T})$ | — | $\sum_t \mathcal{L}(f_\theta, Q_t)$ |
| Meta-Continual Learning | $\{C_{i,1:T}\}_{i=1}^M \sim \mathcal{W}^M$ <br> $S_{i,1:T}, Q_{i,1:T} \sim C_{i,1:T}$ | $f_{\theta_i} = \mathrm{CL}_\phi(S_{i,1:T})$ | $\nabla_\phi \sum_t \mathcal{L}(f_{\theta_i}, Q_{i,t})$ <br> $\forall i < N$ | $\sum_{i=N}^M \sum_t \mathcal{L}(\mathcal{A}_\phi(S_{i,1:T}), Q_{i,t})$ |
| Continual-meta learning | $S_{1:T}, Q_{1:T} \sim C_{1:T}$ | $f_{\theta_t} = \mathcal{A}_\phi(S_{t-1})$ | $\nabla_\phi \mathcal{L}(f_{\theta_t}, S_t)$ | $\sum_t \mathcal{L}(\mathcal{A}_\phi(S_t), Q_t)$ |
| OSAKA | $Q_{1:T} \sim C_{1:T}$ | $f_{\theta_t} = \mathcal{A}_\phi(Q_{t-1})$ | $\nabla_\phi \mathcal{L}(f_{\theta_t}, Q_t)$ | $\sum_t \mathcal{L}(f_{\theta_t}, Q_t)$ |

Table 4: **A unifying framework** for different machine learning settings. Data sampling, fast weights computation and slow weights updates as well as evaluation protocol are presented with meta-learning terminology, i.e., the support set $S$ and query set $Q$. For readability, we omit OSAKA pre-training.

**Meta-continual learning** combines meta-learning and continual learning. A collection of $M$ sequences of contexts is sampled i.i.d. from a distribution over sequences of contexts, $\mathcal{W}^M$, i.e., $\{C_{i,1:T}\}_{i=1}^M \sim \mathcal{W}^M$ and $S_{i,1:T}, Q_{i,1:T} \sim X_{i,1:T} \mid C_{i,1:T}$. Next, the continual learning algorithm, $\mathrm{CL}_\phi$, can be learned using the gradient $\nabla_\phi \sum_t \mathcal{L}(\mathrm{CL}_\phi(S_{i,1:T}), Q_{i,t})$, for $i < N < M$ and evaluated on the remaining sets $\sum_{i=N}^M \sum_t \mathcal{L}(\mathrm{CL}_\phi(S_{i,1:T}), Q_{i,t})$. As in continual learning, the target distribution is fixed.

**Continual-meta learning** considers a sequence of datasets $S_{1:T}, Q_{1:T} \sim C_{1:T}$. At training or continual-learning time, $S_{1:T}$ is both used as a support and query set: $S_t$ is used as the query set and $S_{t-1}$ as the support. Predictions at time $t$ are made using $f_{\theta_t} = \mathcal{A}_\phi(Q_{t-1})$. Since local stationarity is assumed, the model always fails on its first prediction when the task switches. Next, using $l_t = \mathcal{L}(f_{\theta_t}, S_t)$, the learning of $\phi$ is performed using gradient descent of $\nabla_\phi l_t$. The evaluation is performed at the end of the sequence where $\mathcal{A}_\phi$ recomputes fast weights using the previous supports and is tested on the query set, i.e., $\sum_t \mathcal{L}(\mathcal{A}_\phi(S_t), Q_t)$. Similar to meta-learning, continual-meta learning allows for context-dependent targets.

# B  Algorithms

---

**Algorithm 3:** Continual-MAML

---

**1** **Require:** $P(C_{\text{pre}})$, $P(C_{\text{cl}})$: distributions of contexts (or tasks)

**2** **Require:** $\gamma$, $\lambda$: threshold and regularization hyperparameters

**3** **Require:** $\eta$: step size hyperparameter

**4** **Initialize:** $\phi, \theta$: Meta and fast adaptation parameters

**5** **Initialize:** $\eta_\phi$: learnable inner loop learning rate

**6** **Initialize:** $\mathcal{B}$: buffer of incoming data

**7** **while** *pre-training*

**8** $\quad$ Sample batch of contexts (or tasks) $\{C_i\}_{i=1}^B \sim P(C_{\text{pre}})$

**9** $\quad$ **foreach** $C_i$ **do**

**10** $\quad\quad$ Sample data from context $\boldsymbol{x}_i, \boldsymbol{y}_i \sim P(\boldsymbol{x}, \boldsymbol{y} | C_i)$

**11** $\quad\quad$ $\theta_i \leftarrow \phi - \phi_\eta \nabla_\phi \mathcal{L}\big(f_\phi(\boldsymbol{x}_i[:k]), \boldsymbol{y}_i[:k]\big)$

**12** $\quad$ **end**

**13** $\quad$ $\phi \leftarrow \phi - \eta \nabla_\phi \sum_i \mathcal{L}\big(f_{\theta_i}(\boldsymbol{x}_i[k:]), \boldsymbol{y}_i[k:]\big)$

**14** **end**

**15** **Initialize:** current parameters $\theta_0 \leftarrow \phi$

**16** **while** *continually learning*

**17** $\quad$ Sample current context $C_t \sim P(C_{\text{cl}} | C_{t-1})$

**18** $\quad$ Sample data from context $\boldsymbol{x}_t, \boldsymbol{y}_t \sim P(C_t)$

**19** $\quad$ Incur loss $\mathcal{L}\big(f_{\theta_{t-1}}(\boldsymbol{x}_t), \boldsymbol{y}_t\big)$

**20** $\quad$ Virtual model $\tilde{\theta}_t \leftarrow \phi - \phi_\eta \nabla_\phi \mathcal{L}\big(f_\phi(\boldsymbol{x}_t), \boldsymbol{y}_t\big)$

**21** $\quad$ **if** $\mathcal{L}\big(f_{\theta_{t-1}}(\boldsymbol{x}_t), \boldsymbol{y}_t\big) - \mathcal{L}\big(f_{\tilde{\theta}_t}(\boldsymbol{x}_t), \boldsymbol{y}_t\big) < \gamma$

**22** $\quad\quad$ # No context shift detected

**23** $\quad\quad$ Further fine tune the fast parameters

$\quad\quad\quad$ $\theta_t \leftarrow \theta_{t-1} - \phi_\eta \nabla_\theta \mathcal{L}\big(f_{\theta_{t-1}}(\boldsymbol{x}_t), \boldsymbol{y}_t\big)$

**24** $\quad\quad$ Add $(\boldsymbol{x}_t, \boldsymbol{y}_t)$ to buffer $\mathcal{B}$

**25** $\quad$ **else**

**26** $\quad\quad$ # Task boundary detected

**27** $\quad\quad$ Sample training data from buffer $\boldsymbol{x}_{\text{train}}, \boldsymbol{y}_{\text{train}} \sim \mathcal{B}$

**28** $\quad\quad$ Fast adaptation $\theta \leftarrow \phi - \phi_\eta \nabla_\phi \mathcal{L}\big(f_\phi(\boldsymbol{x}_{\text{train}}), \boldsymbol{y}_{\text{train}}\big)$

**29** $\quad\quad$ sample test data from buffer $\boldsymbol{x}_{\text{test}}, \boldsymbol{y}_{\text{test}} \sim \mathcal{B}$

**30** $\quad\quad$ Modulated learning rate $\eta_t \leftarrow \eta g_\lambda \Big( \mathcal{L}\big(f_\theta(\boldsymbol{x}_{\text{test}}), \boldsymbol{y}_{\text{test}}\big) \Big)$

**31** $\quad\quad$ Update Meta parameters $\phi \leftarrow \phi - \eta_t \nabla_\phi \mathcal{L}\big(f_\theta(\boldsymbol{x}_{\text{test}}), \boldsymbol{y}_{\text{test}}\big)$

**32** $\quad\quad$ Reset buffer $\mathcal{B}$

**33** $\quad\quad$ Reset fast parameters $\theta_t \leftarrow \phi - \phi_\eta \nabla_\phi \mathcal{L}\big(f_\phi(\boldsymbol{x}_t), \boldsymbol{y}_t\big)$

**34** $\quad$ $t \leftarrow t + 1$

**35** **end**

---

---

**Algorithm 4:** Continual-MAML w/o Prolonged Adaptation Phase

---

1   **Require:** $P(C_{\text{pre}})$, $P(C_{\text{cl}})$: distributions of contexts (or tasks)
2   **Require:** $\gamma$, $\lambda$: threshold hyperparameters
3   **Require:** $\eta$: step size hyperparameter
4   **Initialize:** $\phi$, $\theta$: Meta and fast adaptation parameters
5   **while** *pre-training*
6     Sample batch of contexts (or tasks) $\{C_i\}_{i=1}^{B} \sim P(C_{\text{pre}})$
7     **foreach** $C_i$ **do**
8       Sample data from context $\boldsymbol{x}_i, \boldsymbol{y}_i \sim P(C_i)$
9       $\theta_i \leftarrow \phi - \phi_\eta \nabla_\phi \mathcal{L}\big(f_\phi(\boldsymbol{x}_i[:k]), \boldsymbol{y}_i[:k]\big)$
10     **end**
11     $\phi \leftarrow \phi - \eta \nabla_\phi \sum_i \mathcal{L}\big(f_{\theta_i}(\boldsymbol{x}_i[k:]), \boldsymbol{y}_i[k:]\big)$
12   **end**
13   **Initialize:** current parameters $\theta_0 \leftarrow \phi$
14   **while** *continually learning*
15     Sample current context $C_t \sim P(C_{\text{cl}}|C_{t-1})$
16     Sample data from context $\boldsymbol{x}_t, \boldsymbol{y}_t \sim P(\boldsymbol{x}, \boldsymbol{y}|C_t)$
17     Incur loss $\mathcal{L}\big(f_{\theta_{t-1}}(\boldsymbol{x}_t), \boldsymbol{y}_t\big)$
18     Reset fast parameters $\theta_t \leftarrow \phi - \phi_\eta \nabla_\phi \mathcal{L}\big(f_\phi(\boldsymbol{x}_t, \boldsymbol{y}_t\big)$
19     **if** $\mathcal{L}\big(f_{\theta_{t-1}}(\boldsymbol{x}_t), \boldsymbol{y}_t\big) - \mathcal{L}\big(f_{\theta_t}(\boldsymbol{x}_t), \boldsymbol{y}_t\big) < \gamma$
20       # No task boundary detected
21       Modulated learning rate $\eta_t \leftarrow \eta g_\lambda \Big(\mathcal{L}\big(f_{\theta_{i-1}}(\boldsymbol{x}_t), \boldsymbol{y}_t\big)\Big)$
22       $\phi \leftarrow \phi - \eta_t \nabla_\phi \mathcal{L}\big(f_{\theta_{t-1}}(\boldsymbol{x}_t), \boldsymbol{y}_t\big)$
23     $t \leftarrow t + 1$
24   **end**

---

## C  Related Work

Our method intersects the topics of continual learning, meta learning, continual-meta learning, and meta-continual learning. For each of these topics, we describe the related work and current state-of-the-art methods.

**Continual learning.**  Given a non-stationary data stream, standard learning methods such as stochastic gradient descent (SGD) are prone to catastrophic forgetting as the network weights adapted to the most recent task quickly cannot perform the previous ones anymore. Many continual learning approaches have been proposed in recent years, which can be roughly clustered into: (1) replay-based methods, (2) regularization-based methods, and (3) parameter-isolation methods. *Replay-based* methods store representative samples from the past, either in their original form (e.g., *rehearsal methods* [59, 26, 63, 2], *constrained optimization* based on those samples [45]), or in a compressed form, e.g., via a generative model [2, 8, 53, 41]. However, those methods require additional storage, which may need to keep increasing when the task sequence is longer. *Regularization-based* or *prior-based* approaches [35, 50, 83] prevent significant changes to the parameters that are important for previous tasks. Most prior-based methods rely on task boundaries. However, they fail to prevent forgetting with long task sequences or when the task label is not given at test time [14, 43]. The third family, *parameter isolation* or *dynamic architecture* methods, attempts to prevent forgetting by using different subsets of parameters for fitting different tasks. This is done either by freezing the old network [81, 67] or growing new parts of the network [40, 66]. Dynamic architecture methods, however, usually assume that the task label is given a test time, which reduces their applicability in real-life settings.

**Meta learning.**  Learning-to-learn methods are trained to infer an algorithm that adapts to new tasks [65]. Meta learning has become central for few-shot classification [58, 78, 52]. A commonly used meta-learning algorithm is MAML [15], which optimizes the initial parameters of a network such that adapting to a new task requires few gradient steps. ANIL [57] is another variation of meta learning that requires only adapting the network's output layer or head to the new tasks. These algorithms leverage gradient descent to learn a feature representation that is common among various tasks, but they are not suitable when the new tasks have a drastic distribution shift from the existing tasks. Despite the limitations of meta-learning methods, they can be adapted to address the challenges of continual learning, as we will describe below.

**Meta-continual learning.**  Since non-stationary data distributions breaks the i.i.d assumption for SGD, it is natural to consider continual learning as an optimization problem where the learning rule learns with non-stationary data. Therefore, some recent works focus on learning a non-forgetting learning rule with meta learning, i.e., meta-continual learning.

In Javed and White [28], the model is separated into a representation learning network and a prediction learning network. The representation learning network is meta learned so that the prediction learning part can be safely updated with SGD without forgetting. In Vuorio et al. [79], a gradient-based meta-continual learning is proposed. The update is computed from a parametric combination of the gradient of the current and previous task. This parametric combination is trained with a meta objective that prevents forgetting.

These approaches are all limited by the fundamental assumption of meta learning that the distribution of the meta testing set matches that of the meta training set. Thus it is not guaranteed that the meta-learned representation or update rule is free of catastrophic forgetting when OoD data is encountered in the future. Despite that, meta-continual learning is actively researched [61, 6].

**Continual-meta learning.**  Recently, several methods emerged that address the continual-meta learning setup. FTML [16] extends the MAML algorithm to the online learning setting by incorporating the follow the leader (FTL) algorithm [22]. FTL provides an $O(\log T)$ regret guarantee and has shown good performance on a variety of datasets. Dirchlet-based meta learning (DBML) [29] uses a Dirchlet mixture model to infer the task identities sequentially.

More relevant to our work, MetaBGD [24] addresses the problem of fast remembering when the task segmentation is unavailable. MOCA [23] extends meta-learning methods with a differentiable Bayesian change-point detection scheme to identify whether a task has changed. Continual-meta learning is now an active research field [46, 5].

### C.1 Contrasting OSAKA and MOCA's framework

In this section, we contrast OSAKA with the recently introduced framework showcasing meta-learning via online changepoint analysis (MOCA) [23]. We are incentivized to discuss these differences because both frameworks can appear similar. Specifically, OSAKA and MOCA's framework represent the tasks or contexts as a hidden Markov chain. However, both settings are fundamentally different and the similarities are superficial. We now highlight their core differences.

**Context-dependent targets** In most CL scenarios including in the MOCA's framework, the joint distribution $p_t(x, y)$ changes through time via the input distribution $p_t(x)$. The target distribution $p(y|x)$ however is fixed (i.e., $p_t(y|x) = p(y|x)$). In other words, in standard incremental CL, new labels still appear even though $p_t(y|x)$ is fixed: they appear via $p_t(x)$ moving its probability mass to new classes.

OSAKA is more general because it allows for drift in the target distribution $p_t(y|x)$ as well. This is achieved through the latent context variable $C$ as detailed in Section 3. In other words, $p_t(y|x) = p(y|x, c_t)$. This is a common scenarios in partially-observable environments [48, 18] or more generally to any case where a prediction depends on the context, e.g. time-series prediction.

**Out-of-distribution tasks** Similar to Javed and White [28], Beaulieu et al. [6], MOCA's framework allows for pre-training. However, all those frameworks test their models on similar data at CL time, i.e., new classes from the same dataset. They thus make strong assumptions about the data distribution that the CL agent will be exposed to at deployment time. This assumption can limite the real-world applicability of current methods.

In OSAKA, pre-training is also allowed. However, at CL time, the model will be tested on OoD data distribution w.r.t the pre-training one (see Section 3. OSAKA thus helps us analyze robustness of algorithms to data distribution(s) outside of the pre-training one. It is thus more aligned with real-life cases of CL.

**Expanding set of labels** In MOCA's framework, all classes are known a priori (see Section B2 in Harrison et al. [23]). They do not allow for an expanding set of labels over time, which is a central idea in CL [35, 45, 59, 14, 2, 4, 68, 28, 11]. MOCA's framework is closer to domain-incremental learning [76], i.e., classes are fixed but new variations can appear within them.

Similarly to standard CL, OSAKA allows for an expending set of labels. Thus, algorithms' capacity to incrementally learn new concepts is studied in OSAKA.

To conclude, the main contribution of Harrison et al. [23] is a new *algorithm*: MOCA. In contrast, OSAKA is a new CL *evaluation framework* aiming to push CL beyond its current limits. We acknowledge that changepoint detection is important for continual learning and refer the readers to [23] for a review of the changepoint detection literature.

# D  Datasets and Baselines

Figure 3: **Benchmarks.** We evaluate our setup on three different benchmarks, each one depicted in one row: Omniglot/MNIST/FashionMNIST, Synbols, and Tiered-ImageNet.

|  | PRE-TRAIN | | CL TIME | | | |
|---|---|---|---|---|---|---|
| MODEL | MAML | ANIL | MAML | SGD | BGD | UM/PAP |
| ONLINE ADAM | × | × | × | √ | × | × |
| FINE TUNING | × | √ | × | √ | × | × |
| BGD [83] | × | × | × | × | √ | × |
| MAML [15] | √ | × | × | × | × | N/A |
| ANIL [57] | × | √ | × | × | × | N/A |
| METABGD [24] | × | × | √ | × | √ | × |
| METACOG [24] | × | × | √ | × | √ | × |
| CONTINUAL-MAML | √ | × | √ | × | × | √ |

Table 5: **Baseline comparison.** Columns 2–3 contain pre-training algorithms. Columns 4–7 show training algorithms at continual learning time. UM and PAP stand for *update modulation* and *Prolonged adaptation phase*, respectively, and are explained in Section 4.

# E  Experiment Details

The procedure followed to perform the experiments in Section 6 is described next in detail. The code to reproduce the experiments is publicly available at `https://github.com/ElementAI/osaka`.

For all experiments, we used a 4-layer convolutional neural network with 64 hidden units as commonly used in the few-shot literature [78, 70, 73, 62]. All the methods were implemented using the PyTorch library [55], run on a single 12GB GPU and 4 CPUs .

## E.1  Hyperparameter search

Hyperparameters were found by random search. During hyperaparmeter search, we allocated the same amount of trials for each method, i.e., each line in the reported Tables. We used Adam [34] for the outer-loop optimization and SGD in the inner (for meta-learning methods). For each trial, we sampled uniformly a method and then sampled hyperparameters uniformly according to the search space defined in Table 7. Each for each hyperparameter trial, we ran two continual learning episodes with different seeds. The seeding impacts the neural net initialization as well as what data stream the algorithm will be exposed to. Whenever the first ran didn't return a cumulative accuracy better than random, we omitted the second run. We allocated equal amount of trials to both non-stationary levels $\alpha \in \{0.90, 0.98\}$. We dedicated a fix amount of compute for each benchmarks and further provide specific details in the rest of this section.

**Omniglot / MNIST / FashionMNIST**   For this benchmark, we allocated a total of 12.5 days of compute. This allowed for 935 trials of which 381 were better than random.

**Synbols**   For this benchmark, we allocated a total of 19.5 days of compute. This allowed for 1,309 trials of which 340 were better than random.

**Tiered-Imagenet**   For this benchmark, we allocated a total of 62 days of compute. We only ran 1 seed per trials which allowed for 934 trials.

For all benchmarks, concerning the runtime per trials, because BGD requires 5 times more compute than SGD, the BGD baseline took approximately five time longer to run than Online ADAM. Similarly, MetaBGD took approximately 5 time longer to run than C-MAML. Moreover, methods with meta-learning took approximately 5 times longer than methods without.

We add the following clarification: we do not need a validation set in OSAKA, as there is no *training error*. Specifically, in the CL episodes, algorithms always make prediction on held-out data.

As for the evaluation runs, the best sets of hyperparameters are used to evaluate the methods on 20 new runs. The algorithms are thus exposed to 20 new CL episodes. For clarification, we do not use the best models found in the hyperparameter-search: we only use the hyperparameters to train and evaluate new models.

| MODEL | $\eta$ | BATCH SIZE | INNER-STEP SIZE | INNER ITERS | FIRST ORDER | MC SAMPLES | $\beta$ | $\sigma$ | $\gamma$ | $\lambda$ |
|---|---|---|---|---|---|---|---|---|---|---|
| ONLINE ADAM | √ | × | × | × | × | × | × | × | × | × |
| FINE TUNING | √ | √ | × | × | × | × | × | × | × | × |
| MAML [15] | √ | √ | √ | √ | √ | × | × | × | × | × |
| ANIL [57] | √ | √ | √ | √ | √ | × | × | × | × | × |
| BGD [83] | √ | × | × | × | × | √ | √ | √ | × | × |
| METABGD [24] | √ | √ | √ | √ | √ | √ | √ | √ | × | × |
| METACOG [24] | √ | √ | √ | √ | √ | √ | √ | √ | × | × |
| CONTINUAL-MAML | √ | × | √ | √ | √ | × | × | × | × | × |
| CONTINUAL-MAML + PRE. | √ | √ | √ | √ | √ | × | × | × | × | × |
| CONTINUAL-MAML + UM | √ | × | √ | √ | √ | × | × | × | √ | × |
| CONTINUAL-MAML + PAP | √ | × | √ | √ | √ | × | × | × | × | √ |

Table 6: **Method's hyperparameters.** $\eta$ is the step-size or outer-step size for meta-learning methods. Batch size is only needed for methods with pre-training. For methods using meta-learning, we searched the inner-step size, the number of inner iterations (inner iters) and the use of the first order approximation of MAML. BGD related hyperparameters, i.e., MC samples, $\beta$ and $\sigma$ are explained in Appendix G.1. $\gamma$ and $\lambda$ are specific of Continual-MAML and operate the update modulation and prolonged adaptation phase mechanisms, respectively. For readability, we omitted 2 hyperparameters related to MetaCOG and refer to the codebase for completeness.

| | | | | | | | |
|---|---|---|---|---|---|---|---|
| $\eta$ | 0.0001 | 0.0005 | 0.001 | 0.005 | 0.01 | | |
| Batch size | 1 | 2 | 4 | 8 | 16 | | |
| Inner-step size | 0.0005 | 0.001 | 0.005 | 0.01 | 0.05 | 0.1 | 0.5 |
| Inner iters | 1 | 2 | 4 | 8 | 16 | | |
| First Order | True | False | | | | | |
| MC Samples | 5 | | | | | | |
| $\beta$ | 0.5 | 1.0 | 10. | | | | |
| $\sigma$ | 0.001 | 0.01 | 0.1 | | | | |
| $\gamma$ | 0.25 | 0.5 | 1.0 | 2.0 | 3.0 | 5.0 | |
| $\lambda$ | 0.25 | 0.5 | 0.75 | 1.0 | 1.25 | 1.5 2.0 2.5 3.0 | |

Table 7: **Hyperparameter search space.**

# F Extra Results

In this section, we provided further results as well as more details about baselines.

## F.1 Omniglot / MNIST / FashionMNIST

In Table 8, we report the full results for the Omniglot / MNIST / FashionMNIST experiment. Contrary to the other experiments, we found that C-MAML pre-training didn't improve results. We thus focus the ablation on C-MAML instead of C-MAML + Pre.

| MODEL | $\alpha = 0.98$ | | | | $\alpha = 0.90$ | | | |
|---|---|---|---|---|---|---|---|---|
| | TOTAL | OMNIGLOT | MNIST | FASHION | TOTAL | OMNIGLOT | MNIST | FASHION |
| ONLINE ADAM | 73.9 ±2.2 | 81.7 ±2.3 | 70.0 ±3.6 | 62.3 ±2.5 | 23.8 ±1.2 | 26.6 ±2.0 | 20.0 ±1.4 | 22.1 ±1.3 |
| FINE TUNING | 72.7 ±1.7 | 80.8 ±2.0 | 68.7 ±2.8 | 59.6 ±3.1 | 22.1 ±1.1 | 25.5 ±1.5 | 18.1 ±1.9 | 19.2 ±1.6 |
| MAML [15] | 84.5 ±1.7 | 97.3 ±0.3 | 80.4 ±0.3 | 63.5 ±0.3 | 75.5 ±0.7 | 88.8 ±0.4 | 68.1 ±0.5 | 56.2 ±0.4 |
| ANIL [57] | 75.3 ±2.0 | 95.1 ±0.6 | 58.7 ±2.9 | 49.7 ±0.3 | 69.1 ±0.8 | 88.3 ±0.5 | 52.4 ±0.6 | 47.6 ±0.9 |
| BGD [83] | 87.8 ±1.3 | 95.1 ±0.5 | 86.9 ±1.1 | 74.4 ±1.1 | 63.4 ±0.9 | 72.8 ±1.2 | 55.9 ±2.2 | 51.7 ±1.3 |
| METACOG [24] | 88.0 ±1.0 | 95.2 ±0.5 | 87.1 ±1.5 | 74.3 ±1.5 | 63.6 ±0.9 | 73.5 ±1.3 | 55.9 ±1.8 | 51.7 ±1.4 |
| METABGD [24] | 91.1 ±2.6 | 96.8 ±1.5 | 92.5 ±1.9 | 77.8 ±3.8 | 74.8 ±1.1 | 83.1 ±1.0 | 71.7 ±1.5 | 61.5 ±1.2 |
| C-MAML | 89.5 ±0.7 | 95.4 ±0.4 | 91.1 ±0.9 | 76.6 ±1.3 | 82.6 ±0.4 | 87.8 ±0.4 | 84.6 ±1.0 | 70.3 ±0.7 |
| C-MAML + KWTO | 92.2 ±0.5 | 97.1 ±0.3 | 94.1 ±0.8 | 80.5 ±1.4 | 84.5 ±0.4 | 88.6 ±0.5 | 86.2 ±0.6 | 74.2 ±0.8 |
| C-MAML + KWTO + ACC. | 92.8 ±0.6 | 97.8 ±0.2 | 93.9 ±0.8 | 79.9 ±0.7 | 83.3 ±0.4 | 89.0 ±0.5 | 84.5 ±0.7 | 71.1 ±0.7 |

Table 8: **Omniglot / MNIST / FashionMNIST experiment**

## F.2 Hyperparameter Sensitivity Analysis

In this section, we analyze the *update modulation* (UM) and *prolonged adaptation phase* (PAP) mechanisms we introduce in C-MAML. Their respective hyperparameters are $\lambda$ and $\gamma$.

We perform the analysis on Synbols for the following reasons: (i) It is harder to solve than the Omniglot benchmark; (ii) Models train faster than Tiered-Imagenet; (iii) It is the only benchmark with an OoD task in which the pre-training data is bestowed a new semantic meaning, i.e., the font classification task.

We analyze the higher non-stationarity setting of $\alpha = 0.98$. setting. This setting puts emphasis on challenging the fundamental i.i.d assumption that CL is interested in solving.

### Update Modulation

We analyze the effect of $\lambda$ parameterizing $g_\lambda : \mathbb{R} \to (0, 1)$. We use $g_\lambda$ to modulate the learning rate proportionally to the loss (see Alg. 2, L23). $\lambda$ provides a smooth interpolation between the behavior of MAML and Continual-MAML. When $\lambda = 0$, Continual-MAML + UM collapses to MAML. When $\lambda = \inf$, Continual-MAML + UM collapses to Continual-MAML. In Figure 4, we show the effect of $\lambda$ on the online cumulative accuracy (same metric as reported elsewhere) which we obtained from our hyperparameter search. Interestingly, all values of $\lambda$ consistently increased the performance of Continual-MAML + UM with respect to MAML and Continual-MAML. This increase is due to two reasons. First, MAML ($\lambda = 0$) cannot accumulate knowledge about the OoD tasks. Second, Continual-MAML (or $\lambda = \inf$) overfits its slow parameters $\phi$ to the current tasks, interfering with previous knowledge too aggressively.

### Prolonged Adaptation Phase

To enable PAP, we need a mechanism to dectect the task boundary (or the context shifts). We propose a simple yet effective context shift detection mechanism which monitors the difference in loss with respect to the previous task and is controlled by a hyperparameter $\gamma$ (Alg. 2, L20). Setting $\gamma$ to high values will increase precision but reduce recall, and vice-versa. In Figure 5 we report precision and recall with respect to multiple values of $\gamma$. We can see that, when tuned appropriately, this mechanism can achieve near-perfect F1 scores, as highlighted by the trials near the top right corner.

The effectiveness of PAP is shown in Figure 6. Specifically, we show that, across all values of $\gamma$, PAP increases the average performance of Continual-MAML. Again, the proposed mechanism is robust to its hyperparameter.

Figure 4: **Update modulation (UM) analysis.** The proposed mechanism is robust to its hyperparameter $\lambda$ and consistently increases average and maximum performance

Figure 5: Precision (y-axis) and Recall (x-axis) for task boundary detection as a function of $\gamma$ (color). **Left:** all trials are plotted, **Right:** trials are grouped by $\gamma$ and the average is reported

Figure 6: **Prolonged adaptation phase (PAP) analysis.** The proposed mechanism increases average and maximum performance.

# G Extra Notes

## G.1 Bayesian Gradient Descent

Bayesian Gradient Descent (BGD) is a continual learning algorithm that models the distribution of the parameter vector $\phi$ by a factorized Gaussian. Similarly to [24] we apply BGD during the continual learning phase. BGD models a the distribution of the parameter vector $\phi$ by a factorized Gaussian $q(\phi) = \prod_i \mathcal{N}(\phi_i | \mu_i, \sigma_i^2)$. Essential motivation behind BGD is that $\sigma$ models the uncertainty of the estimation of the parameter $\phi$. Hence parameters with higher uncertainty should be allowed to change faster than the parameters with lower $\sigma$, which are more important for preserving knowledge learned so far. BGD leverages variational Bayes techniques [21] and introduces an explicit closed-form update rule for the parameters $\mu_i$ and $\sigma_i$:

$$\mu_i = \mu_i - \beta \sigma^2 \mathbb{E}_\epsilon \left( \frac{\partial \mathcal{L}\big(f_{\theta_{t-1}}(X_t), Y_t\big)}{\partial \phi} \right),$$

$$\sigma_i = \sigma_i \sqrt{1 + \left(\frac{1}{2}\sigma_i \mathbb{E}_{\epsilon_i}\left[\frac{\partial \mathcal{L}\big(f_{\theta_{t-1}}(X_t), Y_t\big)}{\partial \phi_i}\epsilon_i\right]\right) -}$$

$$\frac{1}{2}\sigma_i \mathbb{E}_{\epsilon_i}\left[\frac{\partial \mathcal{L}\big(f_{\theta_{t-1}}(X_t), Y_t\big)}{\partial \phi_i}\epsilon_i\right],$$

where the expectations are approximated using Monte Carlo sampling and the re-parametrization trick is used as $\phi_i = \mu_i + \sigma_i \epsilon_i, \epsilon_i \sim \mathcal{N}(0, 1)$.

# H  Q&A

Here you can find reviewers questions and concerns and our answers that we couldn't address in the main part of the paper due to space limitation.

**Pre-training limits the generality of OSAKA and adds computational needs.** We disagree. OSAKA aligns with the deployment of CL systems in real life (Sec. 1 & 3) and it would be more realistic to deploy an agent with some knowledge of the world. Nevertheless, pre-training is not mandatory, although prescribed, and we have a baseline that does use it (C-MAML). Furthermore, it is currently more computationally efficient to learn on i.i.d. data at pre-training than on non-stationary data at CL time and pre-training is a one-time cost compared to CL which is a recurring one.

**Why putting features of different frameworks together is useful for continual learning evaluation?** We unified *and extended* these features to create a more realistic setting than the ones studied in the previous literature. Other frameworks study some of the features in silos but when methods are tested in less realistic settings some methods perform better than they should [12]. See Sec. 6.3 (under dynamic representations) for such an example.

**I think it is strange that MAML performs better in the 0.90 setting.** The reviewer's intuition is right. However, C-MAML needs to predict correctly the context switches otherwise it will get mixed gradients from different tasks. Thus, $\alpha = 0.90$ can be more challenging for methods with dynamic representations when the OoD tasks are not too far from the pre-training ones, as in the Tiered-Imagenet experiment.

**Without task revisits, does $\phi$ stop being suitable for few-shot learning?** It stays suitable because it is still trained with the MAML loss, which optimizes for few-shot learning.