[Reviews · NeurIPS 2020]

Review 1

Summary and Contributions: The authors introduce a new evaluation setting (termed OSAKA) designed for algorithms at the intersection of Meta- & Continual-Learning and delineate several algorithmic settings that have recently proposed in the literature. In addition, the authors introduce a baseline termed Continual-MAML that shows promising performance on he OSAKA benchmark.

Strengths: - The proposed algorithm appears to show promising results based on simple changes to existing work - The underlying data sources are harnessed in an interesting way in the experimental evaluation with OSAKA - The authors clearly describe the distinction between several terms (Meta-, Continual, Continual-Meta, Meta-Continual-Learning) that have caused confusion in the past.

Weaknesses: My main concern with the submission is that the evaluation scenario OSAKA seems too specific and designed primarily for a set of algorithms in between Meta- & Continual-Learning while failing to make a broader argument for other approaches to Continual Learning. While certain aspects of OSAKA are certainly desirable (OOD tasks, Unknown task changes, Online Evaluation) there is a strong assumption made in allowing for Pre-training that may not be adequate in certain CL settings, limiting the generality of OSAKA. Furthermore, it is unclear how aspects such as controllable non-stationarity would be implemented in Reinforcement Learning. Furthermore, I personally feel that if task-revisiting is to be implemented, new OOD tasks should be designed in a way that explicitly re-uses skills that can be learned on a previous problem in a novel setting, instead of merely re-visiting the problem without modification. The problem with this assumption in general is that Catastrophic Forgetting is significantly reduced through an implicit form of replay provided by the environment, making it difficult to tell to which extent catastrophic forgetting is actually a problem of these algorithms. Some questions for the authors: - If task re-visits would be removed, would we expect the pre-trained initialisation \phi to gradually move away from a setting suitable for few-shot learning? - Can the authors think of a setting in which the magnitude of difference between subsequent losses is insufficient to detect a task change? How accurately does the task change detection mechanism predict actual changes? While I understand that a false positive has limited negative impact on the algorithm itself, could an external user rely on the task change signal in case this is required for an application? Finally, how should \gamma be chosen? - Is computational efficiency a suitable claim provided that the proposed algorithm assumes pre-training: a long, costly process that most CL algorithms do not assume? (Related: See also my comment about replay above). - How would you adapt OSAKA to RL?

Correctness: As far as I can tell this seems correct, although I have not checked the empirical methodology and implementation in great details.

Clarity: Unfortunately, there is much room for improvement: - OSAKA appears to be described twice: In the introduction and Section 3. Much needed space could be freed up by sticking to a higher-level introduction and leaving the description of OSAKA exclusively in Section 3. - The terms Continual- & Meta-Learning are used in various combinations that leave the reader confused. I realise that this is partly inherited from prior work and an attempt is made to clear up the confusion, but the primary effect of Section 2on me was tiring and confusing - The description of Meta-Learning is not general enough and appears to exclusively focus on MAML-style algorithms, despite a large literature on other methods. - Algorithm 2: Not a fan of leaving out a large part of the algorithm to save space (14 lines). Comments in the pseudo-code would be helpful. Misc: L144: Typo ("In this setting, models have to if") Figure 1: Shown but not referenced in text

Relation to Prior Work: The related work section almost exclusively focuses on Continual Learning work immediately related to the submission and fails to give a slightly broader overview of the field that should be expected of a NeurIPS publication. Furthermore, some work that the authors heavily rely on (MAML) is simply considered known with little to no technical introduction given to an unfamiliar reader.

Reproducibility: Yes

Additional Feedback: --- Post-rebuttal Update: I thank the authors for their detailed feedback on my initial questions. Personally, I feel that much of our disagreement comes from opposing views on whether we should develop a CL system based on a hunch as to what may be important in the future or based on concrete limitations and impracticalities in current applications (e.g. the discussion on pre-training, task-revisiting). I took this into consideration in my initial assessment and while now agreeing with some answers (and remaining unconvinced with others), overall feel that this score continues to accurately reflect my view on the manuscript.


Review 2

Summary and Contributions: This paper proposes a new framework that encompasses and generalises over continual learning, meta-learning, continual meta-learning and meta-continual learning. In OSAKA pretrained agents are deployed in some environment where their cumulative performance is measured. Agents have no notion of boundaries between tasks which change based on a markov chain. The paper also proposes Continual-MAML, an algorithm suitable for OSAKA, in which parameters change in the inner loop until a large difference indicates a change of task, and then the initialization is also trained to incorporate new knowledge. Continual-MAML is tested on a couple of OSAKA scenarios based on Omniglot, MNIST, FashionMNIST, and Synbols, and experiments show that the proposed algorithm is better than some baselines.

Strengths: - I consider the proposed framework to capture the common desiderata of continual learning and meta-learning, therefore I consider this work relevant to our community - The article add a few tricks on top of MAML yielding a new algorithm which is suitable for the proposed framework (OSAKA) - strong empirical evaluation - well motivated, well placed in the existing literature, and well exposed ideas

Weaknesses: - both the algorithm and the framework are small modifications of previous works. The different assumptions made by OSAKA are not completely novel (e.g. the authors acknowledge MOCA, and other “task-free” / “task agnostic” frameworks), but the authors justify why putting all of them together could be useful for evaluating continual learning algorithms in some domains. - several continual learning papers measured the cumulative performance in task-agnostic setups, maybe without formalising the methodology; therefore in terms of originality the paper doesn't bring much

Correctness: Yes

Clarity: Yes

Relation to Prior Work: Prior work is cited; relevant algorithms are used for baselines

Reproducibility: Yes

Additional Feedback: In table 2, if the bold symbols on each column represent the best score, then for the seventh column (alpha=0.9, previous alphabet), ANIL scores better than C-MAML, but C-MAML’s score is in bold face. If bold symbols emphasize something different, it should be mentioned in the caption.


Review 3

Summary and Contributions: The paper proposed a new evaluation framework for continual learning problem called OSAKA. It also proposed a new algorithm named Continual-MAML which performs competitively in this new framework. The framework addresses a few problems/shortcoming that are observed in modern approaches to continual learning: 1. Revisiting tasks. OSAKA allows for tasks to be revisited. The distribution of tasks follows a Markov chain. This is an important proposition, because in a reality intelligent agents encounter the same or a similar mant time in their lifetime. This suggests to question some of continual learning approaches where new tasks are presented to the model in a prescribed sequence and are never observed again. In such situation if we imagine for a second that a model is a living organism, there is no motivation for it to perform well on previously observed tasks as they are never encountered again. So why should it learn to perform well on them? 2. Supervision over which task is presented right now. Recently the notion of unsupervised continual learning was proposed to characterize a setting where the model is not informed which task it is solving right now. In other words there is no supervision over the task. OSAKA follows this line of thinking and does not provide the task identifier/embedding to the model. This is a much more realistic scenario. 3. OSAKA allows for a different target y to be assigned to the same input x in different contexts. 4. I think the pre-training phase was already present in previously proposed settings. For example MAML algorithm can be considered an algorithm with pre-training phase where the whole meta-learning part is considered pre-training. Nevertheless, it is good that the pre-training phase was highlighted in the paper. 5. Out-of-distribution tasks in OSAKA means that after pre-training the model can encounter a task that is radically different than what was observed in the pre-training phase. I think this is an important aspect. When pre-training tasks and continual learning tasks are similar there is always a doubt of how much performance of the model is due to the similarity between the two and how much due to other factors. 6. Online evaluation. Authors propose that the performance of the model is measured by a cumulative loss summed over the process of learning. It is to be contrasted with the approach where the model is trained in a continual learning fashion and when the training is done it is evaluated on all tasks. In other words authors propose that the performance during the whole process of learning is important rather than the performance of the final model. While I do believe there are settings where the proposed online evaluation is not always the best choice (sometimes we actually care only about the final performance), it is an interesting point of view and definitely a realistic one. The proposed Continual-MAML algorithm operates in two steps. The first step is pre-training which is identical with MAML algorithm. The second step is continual learning. It operates in two variants. The first one is when the model is not observing the out-of-distribution task. In such case the behavior is conceptually similar to MAML with data coming in in a continual learning fashion. The second variant is when an out-of-distribution task is encountered. In such case the initial (slow) parameters \phi are updated towards a solution of the new task. Continual-MAML detects if it has encountered a new task by looking out for a sudden spike in the loss function. If such spike is observed it is assumed that a new task was presented to the model. The experiments section contrasts Continual-MAML with a set of relatively well-known algorithm in the continual learning and meta-learning literature. It seems that the most sophisticated version of Continual-MAML performs very well when compared with these other algorithms. However, I was a bit surprised by results in Table 3. For alpha == 0.98 Continual MAML performs better than other algorithms. While for alpha == 0.90 classical MAML works better. alpha == 0.90 mean more context switching (more switching between tasks), so I think it is a bit strange that MAML performs better in this setting. All the properties of Continual-MAML should allow for quick adaptation to changing distributions of tasks while regular MAML has no extra mechanism to provide such behavior. It would be great if authors can comment on that.

Strengths: The paper clearly states its contributions and goal. It provides a good overview over the current research in continual learning. The authors have identified fundamental problems in the field and proposed a new evaluation framework together with a new simple algorithm which performs well in this setting. I think that the sub-field of continual learning can greatly benefit from more progress both in terms of evaluation benchmarks and new algorithms. Most importantly the new insights have the potential to make the research problems more realistic and closer to natural distribution of tasks/challenges that learning organisms encounter in their lifetime. In this sense the paper makes a contribution in this direction. The novelty of the work is satisfactory. The paper is a very relevant to NeurIPS community because continual learning / lifelong learning remains one of the fundamental unsolved problems when training modern neural networks.

Weaknesses: I have some doubts about the experiments in Table 3. I mentioned some of them in the Summary section. The experimental section could be extended. For example some experiments with reinforcement learning agents would be interesting. Nevertheless, I still think the paper is a valuable contribution to the sub-field of continual-learning and points in the right direction.

Correctness: The claims are support by the results of the Continual-MAML agent.

Clarity: The paper is clear. I encountered only incidental grammar mistakes.

Relation to Prior Work: The prior work is well described and formalized in a unified framework.

Reproducibility: Yes

Additional Feedback: -- Feedback after the rebuttal -- Thank you for responding to my concern about MAML's performance.

[Author Response · NeurIPS 2020]

We thank the reviewers for their thoughtful feedback. We are encouraged they found that: our proposed evaluation framework is interesting, well motivated and captures the important desiderata of continual learning (CL) [**R2**, **R3**, **R4**]; our proposed algorithm C-MAML is suitable for the setting, promising and performs very well [**R2**, **R3**, **R4**]; our empirical evaluation is strong [**R3**]; our proposed unifying framework is well described and clears up confusion caused by prior work or that our work was well placed within the literature [**R2**, **R3**, **R4**]. We are further encouraged by **R4**'s approval on all of OSAKA's features, realistic focus and high relevance for NeurIPS.

**@R2 OSAKA seems too specific while failing to make a broader argument for other approaches to CL** We wholeheartedly disagree. OSAKA's purpose is to align CL research with the *deployment* of autonomous CL systems, e.g., a virtual assistant or a general-purpose robot that would keep on learning about new users and new environments (more examples in Sec. 1 & 3). This scenario encompasses most of the reasons why we study CL. As hypothesised and then shown in the empirical section, other approaches to CL are not well suited to handle the broadness of real-life's requirements or of OSAKA's challenges and thus can not be applied at deployment time.

**@R2 Pre-training limits the generality of OSAKA and adds computational needs.** We disagree. OSAKA aligns with the deployment of CL systems in real life (Sec. 1 & 3) and it would be more realistic to deploy an agent with some knowledge of the world. Nevertheless, pre-training is not mandatory, although prescribed, and we have a baseline that does use it (C-MAML). Furthermore, it is currently more computationally efficient to learn on i.i.d. data at pre-training than on non-stationary data at CL time and pre-training is a one-time cost compared to CL which is a recurring one.

**@R3 Why putting features of different frameworks, e.g., from MOCA, together is useful for continual learning evaluation?** We unified *and extended* these features to create a more realistic setting than the ones studied in the previous literature. Other frameworks study some of the features in silos but when methods are tested in less realistic settings some methods perform better than they should [12]. See Sec. 6.3 (under dynamic representations) for such an example.

**@R3 several continual learning papers already measure the cumulative performance in task-agnostic setups, e.g. MOCA.** We have made a thorough literature review and we haven't found papers that measure cumulative performance apart from concurrent work MOCA. However, we are happy to add these papers if the reviewer includes them in their final review. Furthermore, Appendix B.1 is devoted to contrasting MOCA and OSAKA and describes their differences: context-dependant targets, OoD tasks and the expansion of the set of labels. **R4** (and to some extent **R2**) agrees with the novelty of OSAKA as well as its potential to greatly enhance the field of CL.

**@R4 I think it is strange that MAML performs better in the 0.90 setting.** The reviewer's intuition is right. However, C-MAML needs to predict correctly the context switches otherwise it will get mixed gradients from different tasks. Thus, $\alpha = 0.90$ can be more challenging for methods with dynamic representations when the OoD tasks are not too far from the pre-training ones, as in the Tiered-Imagenet experiment. We added this insight in the paper.

**@R2 How would you adapt OSAKA to RL?** We can replace the current vision tasks by ones from a multi-task RL benchmark such as different tasks in robotics, or change the *contexts* within a standard benchmark, such as different gravity levels in a Mujoco environment. One would control the non-stationarity via the time allocation (or number of episodes) in each task/context.

**@R2 Is task-revisiting an implicit form of replay that reduces catastrophic forgetting?** Yes, because real-life agents encounter the same or similar tasks through their lifetime [**R4**]. Thus, we can build systems that benefit from this natural implicit replay rather then spend computations to constrain models to *always remember all past distributions*.

**@R2 Section 2 on me was tiring and confusing.** As **R2** acknowledged, this is "partly inherited from prior work" and we understand the confusion. It seems however that **R3** and **R4** thought our unifying framework was well described. We would greatly appreciate if the reviewer could expand on this point.

**@R2 Comments in the pseudo-code would be helpful.** Complete algorithms including comments are in Appendix A.

**@R2 Related work is only immediately related to the submission.** Appendix B provides an exhaustive related work.

**@R2 Lack of a technical introduction to MAML.** You can find the MAML algorithm in all the algorithms in Appendix A under the pre-training phase.

**@R2 About the change detection mechanism.** See Figure 4 in Appendix for some results on its accuracy.

**@R2 Without task revisits, does $\phi$ stop being suitable for few-shot learning?** It stays suitable because it is still trained with the MAML loss, which optimizes for few-shot learning.

**@R2,R3** Thanks for pointing out typos, an error in the bolding of results, and a redundant part in the introduction. We have corrected them. We will also use the extra allowed page to increase readability in Sec. 2, answer **R2**'s questions in more details and further justify the unification of OSAKA's features [**R3**].

[Meta-Review · NeurIPS 2020]

This is a borderline paper. The paper has considerable strengths, including a much more realistic continual learning setting for practical applications (please note that there was some disagreement among the reviewers on this, although this area chair sides with the authors), solid positioning with respect to the literature, a method that performs very well in the proposed setting, and strong empirical results. It's main weaknesses are lack of clarity at times, critical details relegated to the appendix, and a lack of discussion/evaluation for how this setting would work in RL (I can easily imagine this, and the authors sketched it in the rebuttal, but it still should have been included in the paper). The authors do need to address these weaknesses before it is ready for publication, since another round of revisions would really benefit the paper. However, I want to commend the authors since I think that this is great work, and look forward to its publication.